# A Study in Pituitary Neuroendocrine Tumors (PitNETs): Real-Life Data Amid Baseline and Serial CT Scans

**DOI:** 10.3390/cancers16203477

**Published:** 2024-10-14

**Authors:** Mihai Costachescu, Oana-Claudia Sima, Mihaela Stanciu, Ana Valea, Mara Carsote, Claudiu Nistor, Mihai-Lucian Ciobica

**Affiliations:** 1Department of Radiology and Medical Imaging, Fundeni Clinical Institute, 022328 Bucharest, Romania; mihaicostachescu@gmail.com; 2PhD Doctoral School of “Carol Davila” University of Medicine and Pharmacy, 010825 Bucharest, Romania; oana-claudia.sima@drd.umfcd.ro; 3Thoracic Surgery Department, “Dr. Carol Davila” Central Emergency University Military Hospital, 010825 Bucharest, Romania; 4Department of Endocrinology, “Lucian Blaga” University of Sibiu, Victoriei Blvd., 550024 Sibiu, Romania; mihaela.stanciu@ulbsibiu.ro; 5Department of Endocrinology, Clinical County Emergency Hospital, 550245 Sibiu, Romania; 6Department of Endocrinology, “Iuliu Hatieganu” University of Medicine and Pharmacy, 400012 Cluj-Napoca, Romania; 7Department of Endocrinology, County Emergency Clinical Hospital, 400347 Cluj-Napoca, Romania; 8Department of Endocrinology, “Carol Davila” University of Medicine and Pharmacy, 0505474 Bucharest, Romania; carsote_m@hotmail.com; 9Department of Clinical Endocrinology V, C.I. Parhon National Institute of Endocrinology, 011863 Bucharest, Romania; 10Department 4–Cardio-Thoracic Pathology, Thoracic Surgery II Discipline, “Carol Davila” University of Medicine and Pharmacy, 0505474 Bucharest, Romania; 11Department of Internal Medicine and Gastroenterology, “Carol Davila” University of Medicine and Pharmacy, 020021 Bucharest, Romania; lucian.ciobica@umfcd.ro; 12Department of Internal Medicine I and Rheumatology, “Dr. Carol Davila” Central Military University Emergency Hospital, 010825 Bucharest, Romania

**Keywords:** pituitary neuroendocrine tumor, incidental finding, imaging, scan, surveillance

## Abstract

**Simple Summary:**

Pituitary neuroendocrine tumors (PitNETs) comprise a large panel of histological types and usually require a multidisciplinary team for diagnosis and management. The most common are clinically and biochemically non-functioning PitNETs, and among them are incidentally found masses of pituitary incidentalomas (PIs), which have an increasing rate of detection currently. PIs are diagnosed in 10% of the population (up to 40% in some studies), and they may be confirmed at any age, mostly by computed tomography (CT) or magnetic resonance imaging (MRI). The rate of PIs growth varies from 5% to 50% depending on the tumor size, but micro-adenomas generally have a lower rate of enlargement. Long-term follow-up is advised, yet standard protocols are not homogenous. Overall, forestalling the tumor’s spontaneous evolution is still an open issue.

**Abstract:**

Non-functioning (NF) accidentally detected PitNETs (PIs) are common findings of CT/MRI scans currently. Data concerning their behavior vary, and some PIs will potentially experience a size change over time that might become clinically relevant. Objective: We aimed to evaluate CT-related PIs diameters following 3 aspects: a cross-sectional analysis based on the age’ groups at first PI diagnosis and on the gender distribution and a longitudinal analysis in PIs with <0.5 cm versus ≥0.5 cm as the largest diameter at baseline. Methods: A retrospective, real-life, multi-centric study in adults with NF micro-PIs was performed. Results: 208 subjects (92.79% females) were included (average age 43.18 ± 12.58 y). The mean largest diameter (between transversal and longitudinal diameters) was 0.55 ± 0.16cm, and 44.71% of the PIs were located on the right part of the pituitary gland. The patients were divided into 10 age-groups (21 to 70 y), and no difference was observed between the mean largest diameters of these groups (*p* = 0.693). Females and males had a similar age at PIs diagnosis (*p* = 0.353), transverse diameter (*p* = 0.910), longitudinal diameter (*p* = 0.229), and PI location (*p* = 0.368). Serial CT scans (2 to 4 per patient) included a median (Q1–Q3) of 20 (12–36) months for the second CT scan, 58 (36–84) for the third CT scan, 78 (53–118) for the fourth CT scan, and a total follow-up between baseline and last CT scan of 40 (13–72) months. The initial largest diameter correlated with the diameter change between the baseline and the last CT (r = −0.575, *p* = 0.000). The largest PI diameter was similar between each serial CT scan (*p* = 0.840). According to the cut-off value of 0.5 cm diameter (for initial largest tumor diameter), group A (N = 78, 37.50%, <0.5 cm) and group B (N = 130, 62. 50%; ≥0.5 cm) had similar age at baseline (43.83 ± 12.72 versus 42.79 ± 12.54 y, *p* = 0.565) and PIs locations (right, left, median). The largest PI diameter remained smaller in group A versus B amid the second CT scan (*p* = 0.000) and the last CT scan (*p* = 0.009). The largest diameter change from the first to the last CT scan showed an increase in group A (median of +0.10 cm, *p* = 0.000) and a decrease in group B (median of −0.01 cm, *p* = 0.002); this diameter change was different in group A versus B (*p* = 0.000). The cumulative probability of tumor-growth-free survival showed different survival functions (log rank *p* = 0.000): group A exhibited a more gradual decrease versus B; at 60 months, the cumulative probability was 0.32 ± 0.08 for group A versus 0.75 ± 0.07 for B. During follow-up, all PIs remained NF, and no hypopituitarism was noted (as limits, we mention that dynamic hypopituitarism testing was selectively performed). Conclusions: NF micro-PIs in adults showed a similar age and sex distribution. During follow-up, PIs with a largest diameter < 0.5 cm increased after a median of 40 months but remained <1 cm, while in PIs with ≥0.5 cm, the largest diameter decreased. This highlights a lower predictability in tumor behavior than expected, particularly in larger micro-PIs that, overall, remained without relevant clinical implications after surveillance.

## 1. Introduction

Pituitary neuroendocrine tumors (PitNETs) embrace a large panel of histological types, and usually, they require a multidisciplinary team for diagnosis and management (that mainly includes endocrine and imaging surveillance or neurosurgery) [1,2]. The most common PitNETs are clinically and biochemically non-functioning, and, among them, incidentally found masses have an increasing rate of detection in countries with a good access to imaging evaluation, such as computed tomography (CT) or magnetic resonance imaging (MRI) [3,4]. For instance, pituitary incidentalomas (PIs) (previously, named “adenomas”) may be found in subjects who underwent various imaging cancer protocols (for malignancies of different non-pituitary origins), suffered a head trauma, or complained of unexplained persistent headache, etc. [5,6,7,8]. 

Generally, PitNETs (pituitary adenomas) prevalence is one out of 1000 people [9], while non-functioning PitNETs that are accidentally found affect approximately 10% of the population (and even 38% to 40% depending on the study design) with no specific age-related incidence as opposed to the adrenal incidentalomas that are more commonly found in aging populations [4,8,10,11,12]. They are usually asymptomatic at first diagnosis and during long standing follow-up, but a 5–10% rate of growth was described, particularly in larger masses (up to 20–50% in voluminous macro-adenomas) [8]. Hence, surveillance is mandatory, especially in early years since initial identification and in tumors that over time showed an increasing diameter according to the imaging-based serial evaluation [4,8]. 

The current study aimed to evaluate CT-related PIs diameters following three aspects: a cross-sectional analysis based on the age’ groups at first PI diagnosis and on the gender distribution, as well as a longitudinal analysis in PIs with <0.5 cm versus those with ≥0.5 cm as the largest diameter at the baseline CT scan. 

## 2. Materials and Methods

This is a retrospective real-life study on individuals who were confirmed with non-functioning PIs (micro-adenomas) by having an initial (native and intravenous contrast) CT scan and an associated clinical and endocrine work-up. The multi-centric data collection was conducted retrospectively according to the approval of the Ethical Committees in each hospital. The inclusion and exclusion criteria are highlighted in Figure 1. 

The confirmation of a non-secreting profile was based on the clinical exam and the hormonal assessments that were performed in each unit. The traditional hormonal panel (baseline blood assays) with regard to the pituitary function was used; of note, some patients had a supplementary dynamic test conducted, other than the baseline blood assays (the dynamic testing data were not analyzed in this study, but they served for confirming the non-functioning pattern in order to include the patient in the present study). 

The studied CT parameters were the two tumor diameters (transversal and longitudinal), and an additional (indirect) diameter, namely the larger one of these two diameters (named “largest diameter” in this analysis), and the specification of the tumor side at the level of pituitary gland (right, left or median). All the multi-centric imaging scans were re-checked by a single radiologist (dr. MC) amid the present analysis. 

The decision of patients’ imaging re-assessment was taken by their current physician, and not according to a standard protocol or timing for each center. Hence, the patients had different serial CT scans during surveillance (from two to four CT scans other than the baseline CT). 

The statistical analysis of the data was performed using Excel 16.78 (Microsoft, Redmond, WA, USA) and SPSS 29.0.2.0 (SPSS, Inc., Chicago, IL, USA). The distribution of each continuous variable was evaluated both empirically and using a Kolmogorov–Smirnov normality test. To illustrate the central tendencies of different variables, mean ± standard deviation (SD) was used for the normally distributed ones and quartiles (Q1, Q2/median and Q3) for those that were not normally distributed. A chi-squared test or Fisher’s exact test was performed to assess statistically significant associations between the categorical variables. A Student’s *t*-test for equality of means was applied to compare the variables with Gaussian distribution and a Mann–Whitney test for the variables with other types of distribution. When comparing the variance across multiple groups, the analysis of the variance test was used (ANOVA test), while, for the non-normally distributed values, a different test was performed (Kruskal–Wallis test) in order to compare the central tendencies of the multiple groups. The linear correlation between two normally distributed variables was established by applying Pearson’s correlation coefficient. For numeric values with non-Gaussian distributions, we applied Spearman’s correlation coefficient. The tumor-growth-free survival curve (Kaplan–Meier curve) provided log-rank test for the statistical significance. Results showing a *p*-value < 0.05 were regarded as statistically significant. 

## 3. Results

The baseline-studied population included 212 patients with PIs (92.92% females). The average age at PI diagnosis was of 42.65 ± 13.05 years. The mean transverse and longitudinal diameters of the tumors were of 0.54 ± 0.17 cm, respectively, of 0.40 ± 0.12 cm, with a mean largest diameter of 0.55 ± 0.16 cm. CT-based localization tumor profile showed that PIs were found as follows: 22.17% in the median part of the pituitary gland, 43.87% on the right side, and 33.96% on the left side (Table 1). 

The endocrine panel confirmed the non-secretor profile in PIs, as mentioned (Table 2). 

After excluding subjects younger than 18 years old, a total of 208 adults were analyzed in this study (92.79% females and 7.21% males), with a mean age of 43.18 ± 12.58 years. CT-scan-based diameters at PI diagnosis had a mean of 0.54 ± 0.17 cm for the transverse diameter, 0.40 ± 0.12 cm for the longitudinal diameter, and 0.55 ± 0.16 cm for the largest diameter. Regarding PI location, 21.63% were in the median part, 44.71% on the right side, and 33.65% on the left side (Table 1). The endocrine assays were confirmatory for non-functional traits (Table 2).

The patients were divided into ten age groups, and the mean for the largest diameter was calculated for each group. No statistically significant difference was observed between the age groups in terms of the largest diameter distribution (*p* = 0.693) (Figure 2). 

Regarding the PI location within the pituitary gland and the age groups, no statistically significant difference was found (*p* = 0.516) (Figure 3). 

When analyzing the cohort by gender distribution, no statistically significant difference was found between females and males in relationship with the age at PIs diagnosis (*p* = 0.353), transverse diameter (*p* = 0.910), longitudinal diameter (*p* = 0.229), or largest CT diameter (*p* = 0.660). No association was found between sex ratio and PI location (*p* = 0.368) (Table 3). 

The patients underwent serial CT scans, as mentioned. The median number of months between the baseline CT scan and the second CT scan was 20, between the initial and the third CT scan was 58, and between the baseline and the fourth CT scan was 78, while between the first and the last CT scan of each patient was 40 (Table 4)

When comparing the baseline (initial)-CT-based largest diameter with the diameter change between the baseline and the last CT, a statistically significant negative correlation was found (r = −0.575, *p* = 0.000). The largest PI diameter was similar between each serial CT scan (according to the second, third, and fourth CT scans) (*p* = 0.840) (Figure 4). 

According to the cut-off value of 0.5 cm diameter (for largest tumor diameter), there was a patients group with less than 0.5 cm diameter (Group A, N = 78, 37.50%) and a group of subjects with tumors having a ≥0.5 cm largest diameter (Group B, N = 130, 62.50%). These two groups had a similar age at first diagnosis (43.83 ± 12.72 versus 42.79 ± 12.54 years, *p* = 0.565). The largest PI diameter was statistically significantly smaller in group A versus group B not only at the baseline (*p* = 0.000) but also amid the second CT scan (*p* = 0.000) and the last CT scan (*p* = 0.009). A similar PI-locations profile within the group A and B was noted (*p* = 0.846) (Table 5). 

The two groups, A and B, had a similar hormonal panel at first evaluation (Table 6). 

The largest diameter change from the first to the last CT scan (median of 40 months, interquartile interval between 13 and 72 months) showed an increase in group A (a median of +0.10 cm) and a decrease in group B (median of −0.01 cm); this diameter change was statistically significantly different in group A versus group B (*p* = 0.000) (Figure 5). 

When analyzing the largest PI diameter within each group (A and B) at initial versus final CT scan, in group A, the PI diameter was statistically significantly larger at the last CT scan when compared to the baseline CT result (*p* = 0.000) and, quite the opposite, in group B, where the largest PI diameter statistically significantly decreased from the first to the last CT (*p* = 0.002) (Figure 6). 

When comparing the cumulative probability of tumor-growth-free survival, these two groups (A and B), as defined by the 0.5 cm cut-off, showed statistically significantly different survival functions (*p* = 0.000): group A exhibited a more gradual decrease compared to group B, and the cumulative probabilities of growth free survival were higher in this group. At 60 months, the cumulative probability of growth-free survival was 0.75 ± 0.07 for group B and 0.32 ± 0.08 for group A (Figure 7, Table 7). 

## 4. Discussion

Since the modern medical era is characterized by an “explosion” of an easy-access to different imaging procedures, such as ultrasound, CT, or MRI, etc., the detection rate of the incidental findings increased for various anatomic locations, including pituitary [13,14,15,16,17]. With regard to the epidemiologic impact, the largest recent study we should mention comes from US population, specifically SEER (Surveillance Epidemiology and End Results) from 2020. The incidence rate of PIs was of 1.53 ± 0.02 per 100,000 people, with a 3 time-increase in the incidence rate from 2004 to 2018. Notably, the incidence rate of pituitary adenomas was 4.28 ± 0.04 per 100,000 persons in 2018, and among these tumors, 24.91% of them were PIs in 2004 versus 42.07% of them in 2018, a statistically significant increase (*p* < 0.001) [18]. Hence, the topic of PI/PitNETs remains very important, and avoiding unnecessary imaging scans might help, but, on the other hand, an adequate timing of re-assessing the patient improves the overall outcome due to an early detection of hormonal anomalies and/or neurologic/ophthalmic damage and a higher chance of their post-operatory correction in surgery candidates [5,6,7,8,9,10,11,12,13,14,15,16,17,18,19]. As opposed to clearly evident conditions (from the clinical perspective) in endocrinology, such as large goiters or full blown Cushing’s syndrome, etc., PIs might remain completely asymptomatic through the entire life span, thus their true natural history is still an open issue currently [20,21,22]. 

### 4.1. Age and Sex-Related Analysis in PIs

In the final analysis, we included 208 adults with a CT confirmation of micro-PIs with a mean age of 43.18 ± 12.58 years. The cohort showed a female predominance of 92.79%. The patients were divided into ten age-based groups (from the age 21 to 70 years), and no difference was observed between the mean largest diameters of these groups (*p* = 0.693). When analyzing the cohort by the gender distribution, no statistically significant difference was found between women and men in relationship with the age at PIs diagnosis (*p* = 0.353), as well as transverse diameter (*p* = 0.910), longitudinal diameter (*p* = 0.229), or largest CT diameter (*p* = 0.660). No association was found between sex ratio and PI location (*p* = 0.368). 

A retrospective multi-centric study conducted by Damilano et al. [23] showed in 67 patients with PIs (67% were females) with a median age of 44 years that 85% of them were macro-PIs, while men had statistically significant larger tumors than women [23]. As mentioned, in our cohort, we had a small number of male adults, but the PIs diameters were similar with the PitNETs in female subjects (of important note, we only included micro-PIs since most literature data agree about this gender distribution, with males more frequently affected particularly in masses larger than 1 cm). Also, Damilano et al. [23] found a positive correlation coefficient between the age at PIs diagnosis and the tumor size (r = +0.31, *p* = 0.001) [23], a correlation that was not statistically significant in our study. Hamblin et al. [24] studied 459 individuals diagnosed with non-functioning micro-adenomas (a median age of 44 years; 66.88% were females, median follow-up of 3.5 years) and showed that age, sex, and baseline diameters of PitNETs were not predictors of the tumor behavior (increase or decrease) [24]. As opposed to our cohort, one out of ten subjects in this analysis had hypopituitarism at baseline MRI scan [24]. Moreover, Freda et al. [25] prospectively studied 269 subjects diagnosed with clinically non-functioning PitNETs, and 48% of them were incidentally detected (PIs). The patients confirmed with PIs versus non-PIs were older, while men diagnosed with PIs and non-PIs were older and had larger tumors when compared to the women harboring PIs and non-PIs, respectively [25]. 

### 4.2. Endocrine Assays (Hormonal Hyper-Activity or Hypo-Activity) in PIs

In this study, none of the tumors proved to associate a secreting profile during follow-up (after a median of 40 months). Initially, we ruled out patients with any hormone excess, but also those with hypopituitarism. Generally, the most common hormonal excess in PitNETs is of prolactin, but most of the non-functioning incidentalomas will remain non-secretors across their life span [26]. Partial or complete single- or multiple-hormone hypopituitarism (most commonly hypogonadism or even a mild form of central adrenal insufficiency) was reported in long-term surveillance of micro-PIs. We did not have enough data to completely rule out a secondary adrenal failure since a routine insulin tolerance test was not performed in every patient, and this might bring a bias. 

Generally, the rate of pituitary insufficiency is higher in non-functioning macro- versus micro-adenomas, affecting 30–45% of the patients, thus the rate of an accidental detection of macro-PIs is reduced to one-third since they present hormonal and/or visual anomalies amid first diagnosis [27]. On the other hand, in micro-adenomas, tumors larger than 5–6 mm have been found at higher risk of hypopituitarism versus smaller PIs, and testing for pituitary hormones deficiencies might start with this cut-off diameter [28]. Apart from the fact that we did not include dynamic hormonal testing, we mention that pituitary endocrine anomalies were not different between group A and B at baseline and amid serial CT scans. 

### 4.3. CT Versus MRI Scans in PIs

At baseline, in this study, the mean largest diameter (between transversal and longitudinal diameters) was 0.55 ± 0.16 cm, and 44.71% of the PIs were located on the right part of the pituitary gland. The initial largest diameter correlated with the diameter change between the baseline and the last CT (r = −0.575, *p* = 0.000). As specified, this was a real-world study conducted in serial CT scans, not MRI or a mixture of those two imaging procedures. Due to radiation and precision considerations, MRI is better than CT to address the issue of PIs, particularly in long-standing surveillance [29] (including in pediatric population [30,31]), but in many centers (as seen in our study), CT is more accessible and reimbursement-free upon hospitalization. 

Generally, the published studies highlighted similar results with concern to PIs/PitNETs diagnosis amid serial CT versus MRI scans, but most studies used MRIs. For instance, one study from 2024 (N = 245 individuals confirmed with asymptomatic non-secretor PIs who were followed for a mean period of 74.2 months based on the clinical exam, respectively, for an average time of 27.3 months based on a MRI scan) showed a size increase in 13.46% of them, and one-third of this specific subgroup underwent neurosurgery [32]. In our cohort, none of the patients required pituitary surgery due to tumor growth and associated compressive elements. 

### 4.4. Tumour Behavior Analysis

The largest PI diameter was similar between each serial CT scan (*p* = 0.840). The largest diameter change from the first to the last CT scan showed an increase in group A (median of +0.10 cm, *p* = 0.000) and a decrease in group B (median of −0.01 cm, *p* = 0.002); this diameter change was statistically significantly different in group A versus group B (*p* = 0.000). 

Generally accepted growth rates of 10% and 25% concern micro-adenomas and macro-adenomas, respectively. But this does not necessarily stand for a major clinical impact, as already pointed out [33]. When comparing the cumulative probability of tumors-growth-free survival, groups showed statistically significantly different survival functions (log rank *p* = 0.000): group A exhibited a more gradual decrease compared to group B; at 60 months, the cumulative probability was of 0.32 ± 0.08 for group A versus 0.75 ± 0.07 for group B.

Our study showed a statistically significant increase in the PIs with the largest diameter less than 0.5 cm (median of 0.41 versus 0.49 cm, *p* = 0.000) upon surveillance for a median of 40 months (interquartile interval: 13–72). As opposite to most of the literature data, which showed a higher risk of tumors growth in larger tumors, we found a reduction in the largest diameter for group B (*p* = 0.002). Of important note, both PI groups remained with a less than 1 cm diameter. Moreover, the fact that an significant rate of PI size decreases during monitoring should not come as a surprise. For example, the prior-mentioned UK Consortium (N = 459) also identified a 7.8% cumulative probability of growth in micro-PIs at 3 years of surveillance and a 14% cumulative probability of tumors reduction upon the same follow-up interval. This may be the natural evolution in such adenomas; alternatively, other factors might influence their diameters or this is a bias of imaging captures [24]. 

This current study did not identify any case of pituitary apoplexy, which is not uncommon for a study of such sample size, noting the rarity of this emergency. Other studies did find this outcome, for instance, Takeshita et al. [34] enrolled 65 individuals with PIs, and 33/65 of them were confirmed as being non-functioning PIs. A relatively high rate of pituitary apoplexy was described in 12% of this subgroup; age, sex, and initial tumors size were not predictors of this complication. However, they found some biochemical anomalies to be prone for this outcome, including liver dysfunction and dyslipidemia [34]. Hamblin et al. [24] also identified one case of pituitary apoplexy during surveillance (1/459 micro-PIs) [24]. Whether the metabolic disturbances associate other genetic and molecular factors at risk for apoplexy in PIs is still an open issue. So far, no isolated risk factor has been regarded as being the single factor responsible for inducing this ailment, while one-third of the patients with pituitary apoplexy have a known precipitating factor [35,36,37,38,39]. 

### 4.5. Limits of the Current Study

This was a real-life retrospective study; hence, we had some limitations. We did not collect data with respect to the stalk deviation in PIs; neither did we include macro-PIs that are more frequently reported to develop secondary hormonal deficiencies and a higher rate of growth with more severe consequences [40]. At baseline, we identified a larger female subgroup, but generally, the incidence was reported equally between females and males in micro-PIs (as mentioned, not all studies agree). For example, the mentioned analysis of SEER database from 2020 showed that PIs versus non-PIs among pituitary adenomas of any type (that are currently named PitNETs, but the study was published in 2021–2022) affected females more often (64% versus 54%, *p* < 0.001), while micro-PIs versus micro-adenomas non-PIs also affected females more frequently (61% versus 13%, *p* < 0.001) [18]. Our cohort did not include tumors with particular imaging traits such as cystic appearance or double incidentalomas since they may embrace a distinct significance from a histological perspective [41,42,43,44,45], and they might bring a bias in the interpretation of long-term tumor behavior, as well as using CT versus MRI scans. 

## 5. Conclusions

In this analysis of more than 200 adults diagnosed with micro-PIs harboring a non-functioning profile, a female predominance was observed (>90%) within a population with an average age of 43.18 years. The patients were divided into 10 age-groups (from 21 to 70 years) and had similar largest diameters. Gender distribution showed similar PI diameters and locations as well as patients’ age at first diagnosis between female and male population. The subjects had 2 to 4 serial CTs, with a total median surveillance of 40 months. The initial largest diameter positively correlated with the diameter change between the baseline and the last CT. During surveillance, no PI proved to associate a secreting profile nor to experience hypopituitarism, and no case of pituitary apoplexy was reported. The largest PI diameter remained statistically significantly smaller in the group with <0.5 cm at baseline versus the group with ≥0.5 cm amid the second and final CT scan, with a largest diameter change in median of +0.1 cm (*p* = 0.000) and −0.01 cm, respectively (*p* = 0.002). When comparing the cumulative probability of tumors-growth-free survival, groups showed statistically significantly different survival functions: group A exhibited a more gradual decrease versus B; at 60 months, the cumulative probability was 0.32 ± 0.08 versus 0.75 ± 0.07 in group A versus B. Group A had an increase in the largest diameter as opposed to B, but none became larger than 1 cm. This study highlights a lower predictability in tumor behavior than expected, particularly in larger micro-PIs that, overall, remained without relevant clinical implications after surveillance. 

## Figures and Tables

**Figure 1 cancers-16-03477-f001:**
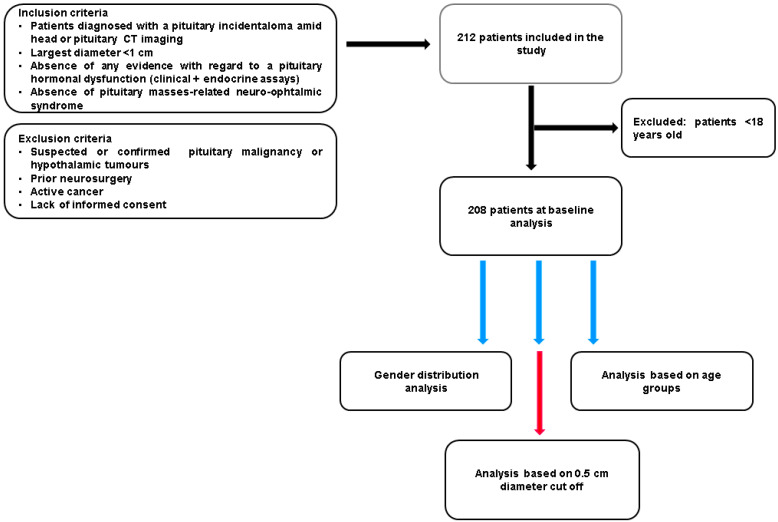
Study protocol and inclusion/exclusion criteria.

**Figure 2 cancers-16-03477-f002:**
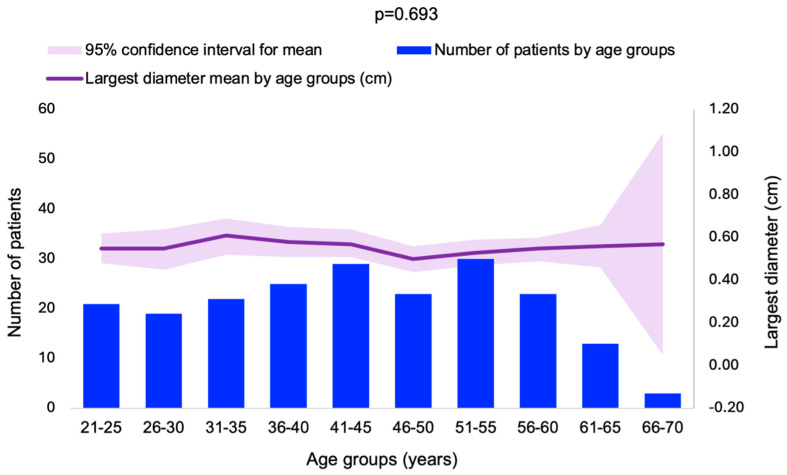
Histogram of frequency distribution and mean largest PI diameter by age groups (N = 208).

**Figure 3 cancers-16-03477-f003:**
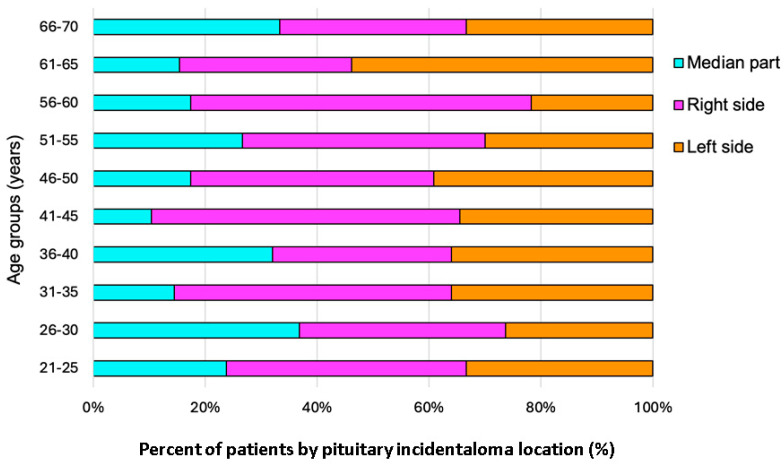
Bar chart showing the frequency of PI location within each age group (N = 208).

**Figure 4 cancers-16-03477-f004:**
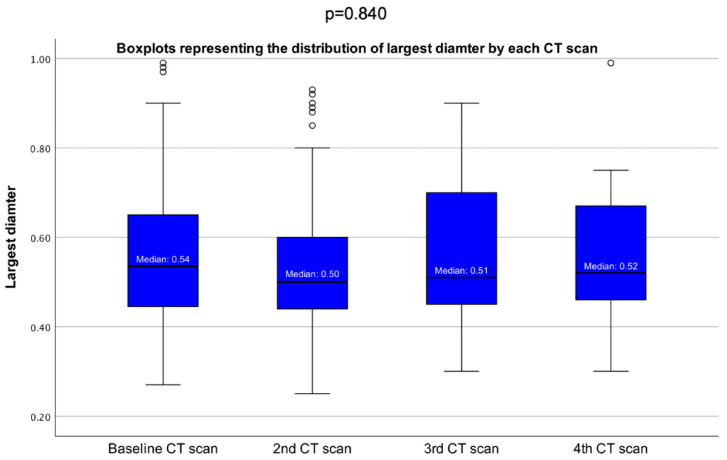
Boxplots showing distributions of the largest tumor diameter by each CT scan.

**Figure 5 cancers-16-03477-f005:**
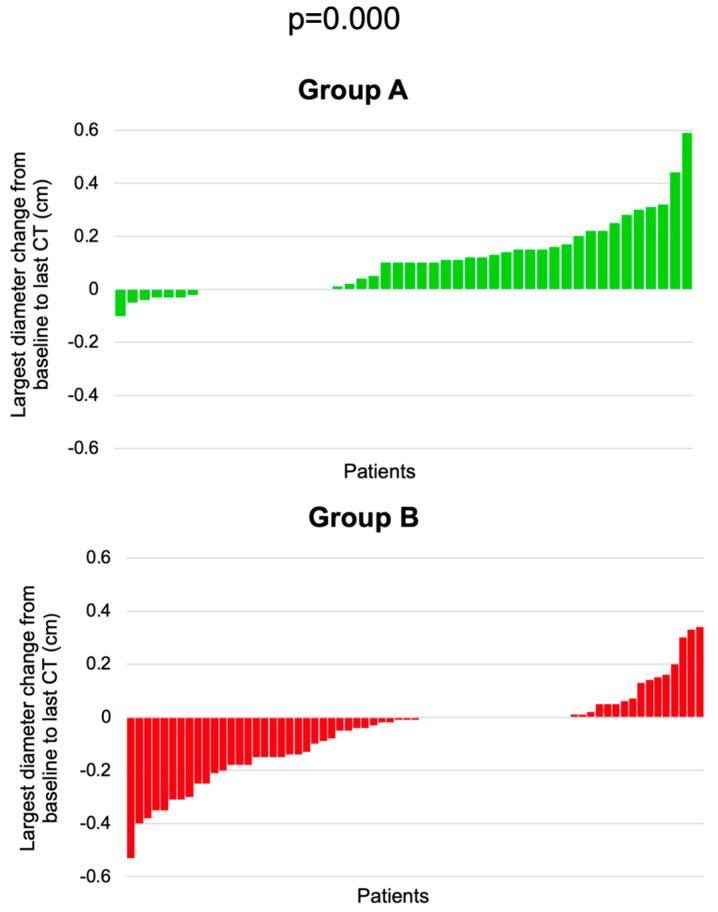
Waterfall plots of largest PI diameter change from the baseline to the last CT scan in group A and group B.

**Figure 6 cancers-16-03477-f006:**
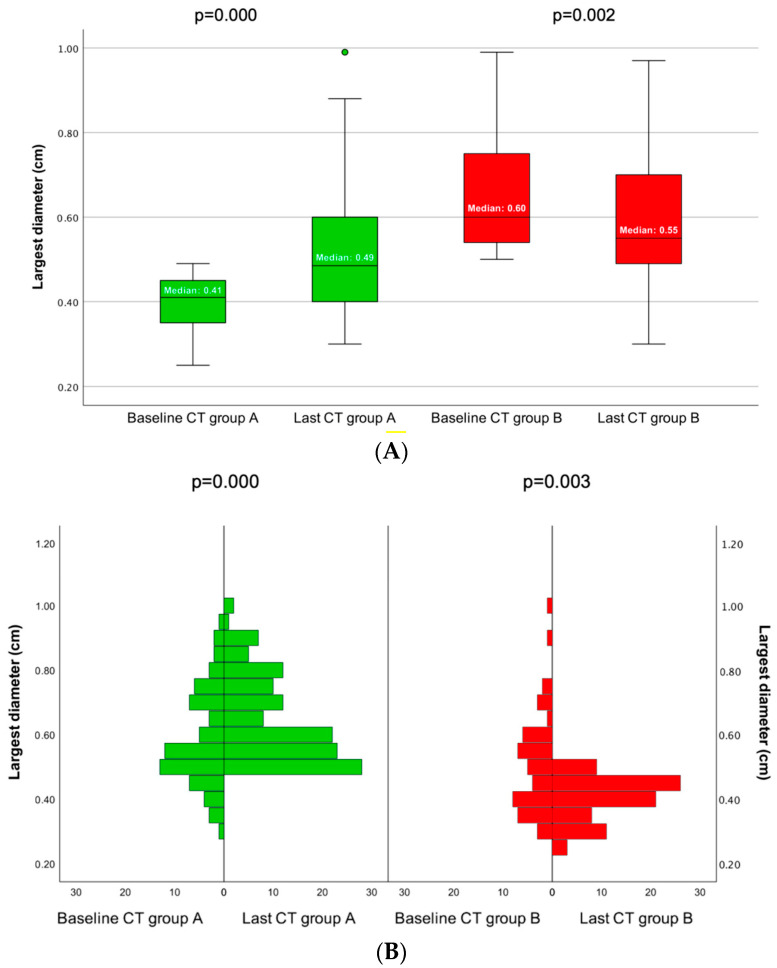
(**A**). Boxplots of the median PI largest diameter at baseline CT versus last CT in group A and group B. (**B**). Frequency distributions of largest PI diameter at baseline CT and last CT in group A and group B (Please see Table 7).

**Figure 7 cancers-16-03477-f007:**
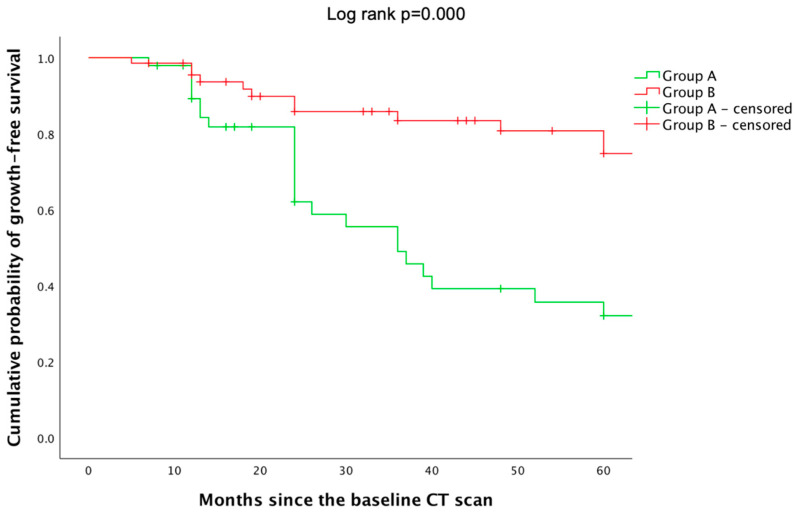
Kaplan–Meier showing the cumulative probability of growth-free survival of group A and group B.

**Table 1 cancers-16-03477-t001:** Characteristics of the baseline-studied population with PIs (N = 208).

Cohort	Females N (%)	Males N (%)	Age (Years) Mean ± SD	TD-CT (cm) Mean ± SD	LD-CT (cm) Mean ± SD	LaD-CT (cm) Mean ± SD	M-CT N (%)	R-CT N (%)	L-CT N (%)
adults (N = 208)	193 (92.79%)	15 (7.21%)	43.18 ± 12.58	0.54 ± 0.17	0.40 ± 0.12	0.55 ± 0.16	45 (21.63%)	93 (44.71%)	70 (33.65%)

Abbreviations: CT = computed tomography; LD = longitudinal diameter; LaD = largest diameter between TD and LD; L-CT = tumor on the left side; M-CT = tumor situated at the median part of the pituitary gland; N = number of patients; R-CT = tumor on the right side; SD = standard deviation; TD = transversal diameter.

**Table 2 cancers-16-03477-t002:** Hormonal blood (baseline) assays in studied population at first PIs diagnosis.

Cohort	GH (ng/mL) Median (Q1, Q3)	IGF1 (ng/mL) Mean ± SD	Prolactin (ng/mL) Mean ± SD	FSH (mIU/mL) Median (Q1, Q3)	LH (mIU/mL) Median (Q1, Q3)	TSH (µIU/mL) Median (Q1, Q3)	ACTH (pg/mL) Mean ± SD	PMC (µg/dL) Mean ± SD
adults (N = 208)	0.30 (0.10, 0.87)	168.61 ± 41.10	8.40 ± 4.76	17.90 (4.97, 60.00)	4.54 (2.62, 30.03)	1.41 (0.97, 2.26)	2 0.24 ± 12.33	12.74 ± 3.60

Abbreviations: ACTH = adrenocorticotropic hormone; FSH = follicular stimulating hormone; GH = growth hormone; IGF1 = Insulin-like Growth Factor 1; LH = luteinizing hormone; PMC = basal plasma morning cortisol; Q = quartile; SD = standard deviation; TSH = thyroid-stimulating hormone.

**Table 3 cancers-16-03477-t003:** Cross-sectional PI analysis according to the gender distribution (N = 208).

Gender Distribution	Number of Patients (%)	Age (Years) Mean ± SD	TD-CT (cm) Mean ± SD	LD-CT (cm) Mean ± SD	LaD-CT (cm) Mean ± SD	M-CT Number of Patients (%)	R-CT Number of Patients (%)	L-CT Number of Patients (%)
Females	193 (92.79%)	43.41 ± 12.43	0.54 ± 0.17	0.40 ± 0.13	0.54 ± 0.17	1 (6.66%)	8 (53.33%)	6 (40.00%)
Males	15 (7.21%)	42.27 ± 14.62	0.53 ± 0.17	0.36 ± 0.09	0.56 ± 0.16	44 (22.80%)	85 (44.04%)	64 (33.16%)
*p*-value		0.353	0.910	0.229	0.660	0.368		

Abbreviations: CT = computed tomography; LD = longitudinal diameter; LaD = largest diameter between TD and LD; L-CT = tumor on the left side; M-CT = tumor situated at the median part of the pituitary gland; R-CT = tumor on the right side; SD = standard deviation; TD = transversal diameter.

**Table 4 cancers-16-03477-t004:** Surveillance imaging analysis: median number of months between the baseline and serial CT scans (N = 208).

Follow-Up Interval between:	Period of Time (Months): Median (Q1, Q3)
Baseline–second CT	20.00 (12.00, 36.00)
Baseline–third CT	58.00 (36.00, 84.00)
Baseline–fourth CT	78.00 (53.50, 118.50)
Baseline–last CT	40.00 (13.00, 72.00)
second–third CT	24.00 (16.25, 48.00)
second–fourth CT	48.00 (36.00, 84.00)
third–fourth CT	32.00 (23.00, 51.00)

**Table 5 cancers-16-03477-t005:** Characteristics of the patients in group A (largest PI diameter of <0.5) and B (largest PI diameter ≥ 0.5 cm) according to the initial CT scan: baseline and serial CT scan analysis.

Studied Group	Number of Patients (%)	Age (Years) Mean ± SD	LaD-CT (cm) (Baseline CT) Mean ± SD	LaD-CT (cm) (2nd CT Scan) Mean ± SD	LaD-CT (cm) (3rd CT Scan) Mean ± SD	LaD-CT (cm) (4th CT Scan) Median (Q1, Q3)	LaD-CT (cm) (Last CT Scan) Mean ± SD	LaD-CT Change between Baseline-Last CT Scan (cm) Median (Q1, Q3)	M-CT Patients (%)	R-CT Patients (%)	L-CT Patients (%)
group A	78 (37.50%)	43.83 ± 12.72	0.40 ± 0.06	0.47 ± 0.12	0.55 ± 0.18	0.56 (0.41, 0.68)	0.51 ± 0.15	0.10 (0.00, 0.16)	18 (23.08%)	34 (43.59%)	26 (33.33%)
group B	130 (62.50%)	42.79 ± 12.54	0.65 ± 0.13	0.58 ± 0.15	0.57 ± 0.14	0.50 (0.47, 0.66)	0.59 ± 0.15	−0.01 (−0.15, 0.00)	27 (20.77%)	59 (45.38%)	44 (33.85%)
*p*-value		0.565	0.000	0.000	0.694	0.800	0.009	0.000	0.846		

Abbreviations: CT = computed tomography; LaD = largest diameter between transverse and longitudinal diameter; L-CT = tumor on the left side; M-CT = tumor situated at the median part of the pituitary gland; R-CT = tumor on the right side; SD = standard deviation.

**Table 6 cancers-16-03477-t006:** Baseline endocrine assays in the studied population at first PIs diagnosis according to the group A and group B.

Studied Group	GH (ng/mL) Median (Q1, Q3)	IGF1 (ng/mL) Mean ± SD	Prolactin (ng/mL) Mean ± SD	FSH (mIU/mL) Median (Q1, Q3)	LH (mIU/mL) Median (Q1, Q3)	TSH (µIU/mL) Median (Q1, Q3)	ACTH (pg/mL) Median (Q1, Q3)	PMC (µg/dL) Mean ± SD
group A	0.33 (0.08, 0.61)	159.49 ± 3 0.62	9.21 ± 5.35	39.36 (21.43, 67.24)	22.69 (3.08, 53.17)	1.29 (0.93, 1.82)	16.31 (7.31, 22.13)	11.91 ± 2.61
group B	0.32 (0.10, 0.96)	178.23 ± 51.15	8.05 ± 5.36	5.96 (4.91, 36.70)	3.58 (2.50, 19.44)	1.62 (0.98, 2.59)	22.07 (13.62, 28.84)	13.33 ± 4.12
*p*-value	0.698	0.222	0.342	0.102	0.143	0.154	0.182	0.231

Abbreviations: ACTH = adrenocorticotropic hormone; FSH = follicular stimulating hormone; GH = growth hormone; IGF1 = Insulin-like Growth Factor 1; LH = luteinizing hormone; PMC = basal plasma morning cortisol; Q = quartile; SD = standard deviation; TSH = thyroid stimulating hormone.

**Table 7 cancers-16-03477-t007:** Cumulative probabilities of growth-free survival.

Time since Baseline CT Scan	Cumulative Probability of PI Growth-Free Survival	
	Group A	Group B
12 months	0.89 ± 0.05	0.96 ± 0.03
24 months	0.62 ± 0.08	0.86 ± 0.05
36 months	0.49 ± 0.09	0.83 ± 0.05
48 months	0.39 ± 0.09	0.81 ± 0.06
60 months	0.32 ± 0.08	0.75 ± 0.07

## Data Availability

All data generated or analyzed during this study are included in this article. Further enquiries can be directed to the corresponding authors.

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
