# Peer review of "A Study in Pituitary Neuroendocrine Tumors (PitNETs): Real-Life Data Amid Baseline and Serial CT Scans"

_cancers, 2024, doi:10.3390/cancers16203477_

Round 1

Reviewer 1 Report

Comments and Suggestions for Authors

The study by Mihai Costachescu et al. titled "A Study in pituitary neuroendocrine tumors (PitNETs): real-life data amid baseline and serial scans 3" is a study that describes and analyzes through CT scan of the skull to visualize pituitary, the presence of pituitary micro incidentalomas and then describe their general characteristics such as gender, age and size of the micro PIs, they divide them according to their maximum diameter and after survival analysis they observe that in their results there are some outcomes that are not consistent with previous literature. It is known that pituitary incidentalomas in their natural history are more likely to increase their growth in larger micro-PIs, specifically >0.6cm, with clearly established risks as the initial size of pituitary microincidentalomas increases. Important limitations of having had MRI studies in all patients since this imaging study is more sensitive in small lesions, and reduces some implicit classification biases.

1.- Neuroendocrinology 2012;96:333–342

2.-European Journal of Endocrinology 165 739–744

Comments on the Quality of English Language

no comments

Author Response

Response to Review 1 Comments

Dear Reviewer,

Thank you very much for your time and your effort to review our manuscript.

We are very grateful for providing your valuable feedback on the article.

Here is our response and related amendment that has been made in the manuscript according to your review (marked in yellow color).

Comments and Suggestions for Authors: The study by Mihai Costachescu et al. titled "A Study in pituitary neuroendocrine tumors (PitNETs): real-life data amid baseline and serial scans 3" is a study that describes and analyzes through CT scan of the skull to visualize pituitary, the presence of pituitary micro incidentalomas and then describe their general characteristics such as gender, age and size of the micro PIs, they divide them according to their maximum diameter and after survival analysis they observe that in their results there are some outcomes that are not consistent with previous literature. It is known that pituitary incidentalomas in their natural history are more likely to increase their growth in larger micro-PIs, specifically >0.6cm, with clearly established risks as the initial size of pituitary microincidentalomas increases.

Thank you very much. Indeed, some of the presented data were not consistent with prior literature, mostly amid a real-life study. Thank you

Important limitations of having had MRI studies in all patients since this imaging study is more sensitive in small lesions, and reduces some implicit classification biases.

Thank you very much. We introduced at Discussion the importance of the MRI rather than CT and also at limitations sub-section. Yet, in many centers the patients might have a CT scan free of reimbursement or the initial medical or surgical condition (that finally proved to be unrelated to the PIs diagnosis) is suitable to be explored amid using CT scan rather than a MRI exam. Moreover, some patients have MRI contraindications (e.g. implantable devices or other clips, etc.), hence, CT might prove a useful tool. Thank you

1.- Neuroendocrinology 2012;96:333–342

2.-European Journal of Endocrinology 165 739–744

Thank you very much. We followed your recommendations. Thank you

Comments on the Quality of English Language: no comments

Thank you very much. We really appreciate it!

Reviewer 2 Report

Comments and Suggestions for Authors

I would remove the reference to "real - life" from the title of the article and the text, since the use of CT is not universally recommended for studying the hypothalamic-pituitary region. The greater availability of the study in a single medical institution, in my opinion, as well as the restrictions applied (excluding incidentalomas larger than 1 cm, cystic adenomas, not using contrast), indicate a shift in the sample from real life.

Minor noted edits:

- Line 179 - in the caption to Figure 3, it is necessary to change pacients to patients

- Line 238 - table number 7

Comments on the Quality of English Language

 Minor editing of English language required.

Author Response

Response to Review 2 Comments

Dear Reviewer,

Thank you very much for your time and your effort to review our manuscript.

We are very grateful for your insightful comments and observations, also, for providing your valuable feedback on the article.

Here is a point-by-point response and related amendments that have been made in the manuscript according to your review (marked in yellow color).

Comments and Suggestions for Authors: I would remove the reference to "real - life" from the title of the article and the text, since the use of CT is not universally recommended for studying the hypothalamic-pituitary region.

Thank you very much for pointing this aspect. We respectfully mention that the real-life settings capture the essence of the study since many patients underwent CT scan for unrelated conditions (to PIs) that are suitable for a CT scan (rather than a MRI) or they have contraindications to a MRI or the MRI is not free of reimbursement in some centers (for some diseases) as it is for a CT scan. In accordance to your observations, a sub-section of Discussion is dedicated to MRI versus CT scan aspects in PIs and we mentioned at limitations the real-life study design. Thank you

The greater availability of the study in a single medical institution, in my opinion, as well as the restrictions applied (excluding incidentalomas larger than 1 cm, cystic adenomas, not using contrast), indicate a shift in the sample from real life.

Thank you very much. At Methods we specified “All the multi-centric imaging scans were re-checked by a single radiologist (dr. MC) amid the present analysis“.

At limits of the study we specified: “Our cohort did not include tumors with particular imaging traits such as cystic appearance or double incidentalomas since they may embrace a distinct significance from a histological perspective”.

All patients had a native and an intravenous contrast CT scan. We thank you very much for pointing this aspect and we added this useful information to the Methods section.

Thank you

Minor noted edits:

- Line 179 - in the caption to Figure 3, it is necessary to change pacients to patients

- Line 238 - table number 7

Thank you very much. We corrected it.

Comments on the Quality of English Language: Minor editing of English language required.

Thank you very much. We corrected it.

Thank you very much

Reviewer 3 Report

Comments and Suggestions for Authors

The paper is interesting and topic developed quite well.

Some suggestions/corrections:

1) Ln 48 “with NF mi- 48 cro-PIs with” is not correct.

2) The abstract is too long and it must be shorted. It should be a total of about 250 words and results must summarize the article’s main findings.

3) If you excluded subjects younger than 18 years old for your analysis, I suggest to remove their confounding data about baseline patients in tables 1 and 2; then table 1 and 2 should be placed after ln 170.

4) In Fig 3 “pacients” is not correct.

5) Tables 1-2-3-5 are very full: it is advisable to use abbreviations whenever possible (e.g. “number of patients”)

6) I encourage authors to add some emblematic figures with captions showing the pituitary neuroendocrine tumours in baseline and serial CT scans.

7) The conclusion is too long and it must be shorted. It should contain the main information obtained from the processing of the collected data.

Comments on the Quality of English Language

Minor editing of English language required.

Author Response

Response to Review 3 Comments

Dear Reviewer,

Thank you very much for your time and your effort to review our manuscript.

We are very grateful for your insightful comments and observations, also, for providing your valuable feedback on the article.

Here is a point-by-point response and related amendments that have been made in the manuscript according to your review (marked in yellow color).

Comments and Suggestions for Authors

The paper is interesting and topic developed quite well.

Thank you very much. We really appreciate it!

Some suggestions/corrections:

Ln 48 “with NF mi- 48 cro-PIs with” is not correct.

    Thank you very much. We corrected it.

The abstract is too long and it must be shorted. It should be a total of about 250 words and results must summarize the article’s main findings.

Thank you very much. We revised it.

If you excluded subjects younger than 18 years old for your analysis, I suggest to remove their confounding data about baseline patients in tables 1 and 2; then table 1 and 2 should be placed after ln 170.

Thank you very much. According to your recommendations, we removed the data that included patients younger than 18 years old from table 1 and 2 and moved these tables after line 170. Thank you

In Fig 3 “pacients” is not correct.

Thank you very much. According to your recommendation, we corrected it.

Tables 1-2-3-5 are very full: it is advisable to use abbreviations whenever possible (e.g. “number of patients”)

Thank you very much. We corrected it and provide the abbreviations after the table.

I encourage authors to add some emblematic figures with captions showing the pituitary neuroendocrine tumours in baseline and serial CT scans.

Thank you very much. We respectfully mention that this study was intended to capture the essence of the cohort, not case reports. Thank you

The conclusion is too long and it must be shorted. It should contain the main information obtained from the processing of the collected data.

Thank you very much. We revised it.

Comments on the Quality of English Language: Minor editing of English language required.

Thank you very much. We corrected it.

Thank you very much